# The Effects of Playground Interventions on Accelerometer-Assessed Physical Activity in Pediatric Populations: A Meta-Analysis

**DOI:** 10.3390/ijerph19063445

**Published:** 2022-03-15

**Authors:** Christopher D. Pfledderer, Sunku Kwon, Ildiko Strehli, Wonwoo Byun, Ryan D. Burns

**Affiliations:** 1Department of Exercise Science, University of South Carolina, Columbia, SC 29208, USA; chris.pfledderer@sc.edu; 2Department of Health & Kinesiology, University of Utah, Salt Lake City, UT 84112, USA; sunku.kwon@utah.edu (S.K.); ildiko.strehli@utah.edu (I.S.); won.byun@utah.edu (W.B.)

**Keywords:** adolescents, children, environment, physical activity, schools

## Abstract

Playgrounds are designed to be a safe, enjoyable, and effective means to promote physical activity in children and adolescents. The purpose of this study was to conduct a meta-analysis to determine the effectiveness of playground interventions for improving accelerometer-assessed ambulatory moderate-to-vigorous physical activity (MVPA) and to identify common aspects of playground interventions that may be beneficial to promote behavior change. An internet database search was performed. The final analyzed sample of studies was obtained from several criteria, including being a playground-based intervention targeting children or adolescents, having a control or comparison group, having an accelerometer-assessed MVPA outcome target variable, and reporting of the mean difference scores’ variability. A random-effects model meta-analysis was employed to obtain pooled effect sizes. Ten studies (*n* = 10) were analyzed from the internet search. The weighted pooled effect (Hedges’ g) across all studies was Hedges’ g = 0.13, 95% CI: 0.02–0.24, *p* = 0.023. There was moderate study heterogeneity (*I*^2^ = 55.3%) but no evidence for publication bias (*p* = 0.230). These results suggest that school-based playground interventions have a small effect on increasing accelerometer-assessed MVPA within the pediatric population. The playground should still be an environmental target during school or community-based interventions aimed at providing opportunities to promote MVPA.

## 1. Introduction

The favorable link between physical activity, health, and wellness in youth has been well established in the literature. Some of the outcomes associated with increased physical activity in this population include improvements in body composition and health-related physical fitness [1], higher grade point average [2], increased wellness, and improved mental health [3]. Despite this mounting evidence, many developed nations’ youth are not meeting recommended levels of physical activity. In the UK and the US, less than 25% of the youth population meets the recommended duration of moderate-to-vigorous physical activity as recommended by various health agencies (60 min) [4,5]. One promising venue for MVPA promotion and interventions among children and adolescents is the playground setting, as interventions specifically focused on this location have shown effectiveness for improving levels of MVPA among youth participants [6].

A playground is typically defined as an area that has been specifically designated for play or recreation. Usually, these areas are outdoors and contain play equipment including slides, swing sets, and jungle gyms, as well as defined areas for other types of recreation such as baseball diamonds, hopscotch, and/or foursquare. Many playgrounds are also connected to a school, allowing children and adolescents to use them during recess and other break times, although public playgrounds not associated with schools are common as well.

The playground is a logical location for physical activity interventions to occur, especially playgrounds connected to schools, as altering aspects of the physical environment to increase activity is an essential part of wellness promotion in youth [7]. Further, the playground is an area that is specifically designed to promote physical activity and can be an essential resource where children are able to play safely [8]. While there is little research on the effect of playgrounds on other beneficial outcomes besides physical activity [9], some studies have found that recess (a time when children have access to playgrounds) can have a positive effect on attention [10], and teachers have reported children being more social, creative, and resilient when playgrounds are improved with equipment [11]. The playground is also a space where youth can develop socially. In fact, several studies have explored associations between playground engagement and outcomes such as social network inclusion [12], social interaction [13], and perceived “restorativeness” [14], with positive results. Clearly, the playground should be a location of interest when designing interventions aimed at promoting physical activity in children and adolescents.

During playground activity, an objective estimation of MVPA may help inform physical activity programs with respect to the reporting of adequate ambulatory movement duration and intensity. Accelerometers are frequently used activity monitors that assess physical activity both reliably and with a high degree of construct validity [15,16]. Accelerometry-assessed measurement can provide valid physical activity scores and valid scores of the amount of time spent in sedentary behaviors in a variety of settings [17,18]. Therefore, accelerometry-assessed MVPA can yield important information that may aid in understanding the effect of playground-based interventions.

While there have been numerous studies exploring the beneficial aspects of play areas on movement behaviors in children and adolescents using accelerometry-based assessment methods, there is a paucity of meta-analyses aiming to test pooled effects. Escalante et al. [19] published a paper exploring the influence of playground design on physical activity. The authors found that playground markings and equipment did not increase activity, whereas interventions on physical structures in playgrounds increased physical activity during recess. While results from this systematic review add important information to the literature, the authors included studies that used both accelerometer-based and heart-rate-based assessment and did not use meta-analysis to test pooled effects and between-study heterogeneity. In recent years, studies of play areas have taken an even broader approach, exploring relationships between physical activity interventions during recess [20,21] or the classroom [22]. Erwin et al. [20] found a significant and moderate pooled effect of recess interventions on seven domains related to physical activity, including age, gender, intervention type, intervention duration, duration of physical activity intervention session, outcome measures, and study regions.

To identify aspects of the environment where programs can be beneficial in promoting physical activity in pediatric populations, it is important to consider studies that have focused on specific locations, such as playgrounds. Furthermore, it is important to consider studies that have used similar assessment techniques, as the collective results of these studies can be more easily used to make future recommendations that might guide playground-intervention design. Given the fact that playgrounds are areas where youth frequently engage in active play, it is important to gather information on the underlying relationships between facets of play areas with potential associations with MVPA. However, no study has focused on the pooled relationship between playground interventions and accelerometry-assessed MVPA in children and adolescents. The aims of this study were to (1) assess pooled effect sizes of playground-based studies on accelerometry-based measurements of MVPA among children and adolescents and (2) determine the facets of the studies that had a favorable influence on MVPA outcomes.

## 2. Materials and Methods

### 2.1. Study Selection

An online search used several keywords—namely, “children”, “youth”, “playgrounds”, “accelerometer”, “physical activity”, and “sedentary behavior”—was performed; over 254 studies were obtained on the topic between 1 January 2000–31 December 2021 after duplicate removal. MEDLINE/PubMed, Scopus, EMBASE, and PsycINFO databases were used. The Population, Intervention, Comparison, and Outcome-PICO” framework was employed with “OR” and “AND” Boolean operators. Studies were then selected based on our inclusion criteria, which included playground-based intervention studies that used accelerometer-assessed MVPA as a target variable, interventions with youth (<18 years old) as participants, and studies that had a control/comparison group. We eliminated studies if not enough information was provided that was needed to calculate the standardized mean differences or effect sizes. Alternative interventions (e.g., nature-based interventions) were not included unless a playground was the primary locus of intervention. This study’s selection protocol yielded 10 studies that were used in the meta-analysis. These procedures are outlined in Figure 1.

### 2.2. Physical Activity Assessment

The published manuscripts in this review all utilized accelerometers. Both waist- and wrist-worn methodology were considered. To attenuate between-study heterogeneity, only outcomes reporting MVPA were considered. Assessment units considered were MVPA (in minutes/day) and the percentage of accelerometer wear time in MVPA. A variety of count processing algorithms were considered to keep the final sampling pool as large as possible. All results were standardized to account for these assessment differences.

### 2.3. Standardized Mean Differences

We calculated standardized mean differences (SMDs) from experimental group sample sizes, mean differences based on changes from before and after the intervention on accelerometer-assessed MVPA, and mean difference variability (standard deviations). SMDs, rather than absolute mean differences, were used because of the difference in the types of assessment metrics employed across the 10 selected studies. These data were then entered into a statistical software program for analysis.

### 2.4. Statistical Analysis

This meta-analysis examined if there was any pooled effect of playground interventions for improving MVPA. The DerSimonian and Laird random-effects model was employed. Studies were weighted by inverse variance, which included both within-study and between-study heterogeneity. Hedges’ g quantified the study and pooled effects. A forest plot with corresponding 95% confidence intervals was used to visually show the results of the meta-analysis. Hedges’ g scores were considered small if g < 0.20, medium if g = 0.50, and large if g ≥0.80 [23]. Cochran’s Q test quantified study heterogeneity along with the *I*^2^ statistic. An alpha level set at *p* ≤ 0.10 for Cochran’s Q and the *I*^2^ statistic determined low if *I*^2^ < 50%, moderate if *I*
^2^ = 50–75%, and large if *I*^2^ > 75% [24]. Publication bias was assessed using funnel plots. The funnel plots displayed the standardized mean differences (SMDs) on the x-axis and the standard error of the SMDs on the y-axis. Egger’s linear regression test assessed publication bias [25]. Post hoc sensitivity analyses were conducted by removing a single study for the analysis across 10 iterations to determine if Hedges’ g scores would be different. Tabular results were reported for the sensitivity analysis. Alpha level was set at *p* < 0.05, and all analyses were conducted using Stata version 17.0 statistical software package (StataCorp., College Station, TX, USA).

## 3. Results

### 3.1. Descriptive Statistics

Table 1 summarizes all of the selected studies [26,27,28,29,30,31,32,33,34,35]. In total, 3502 participants were included in the review [26,27,28,29,30,31,32,33,34,35]. Due to lost-to-follow-up or missing accelerometer data, 3247 participants were analyzed (intervention group = 1787; control group = 1460; 87% of original sample). Two studies were from Australia [29,31], one study was from Belgium [30], two studies were from France [26,28], one study was from New Zealand [32], and four studies were from the United Kingdom [27,33,34,35]. Mean age in the individual studies ranged from 5.3 ± 0.4 years to 8.8 ± 0.5 years.

### 3.2. Intervention Characteristics

All of the reviewed papers included school-based programs [26,27,28,29,30,31,32,33,34,35]. Intervention duration ranged from 4 school days to 2 years. Four interventions employed the sporting playground zonal design [26,28,34,35], four studies used novel playground markings [26,28,30,35], two studies added new small equipment to the playground [27,30], two studies added recycled materials with no obvious play use [29,31], two studies involved novel structures [33,35], two studies involved teacher–parent workshops [29,31], one study involved playground action plans [32], and one study involved free play [27]. Most studies involved using a combination of methods to elicit physical activity participation [26,27,28,29,30,31,33,34,35].

### 3.3. Meta-Analysis Findings

Figure 2 visually illustrates the results of the meta-analysis. Hedges’ g scores ranged from −0.16 to (negative small effect) to 0.27 (positive small effect). Two studies reported negative effect sizes, suggesting lower MVPA after the playground intervention [32,33]; the remaining studies showed a positive trend or positive effect for improving MVPA. Using a random-effects meta-analysis approach [36], the pooled Hedges g = 0.13, (95% CI: 0.02–0.24), which was a small effect size. However, the z-statistic = 2.28 (*p* = 0.023), indicating statistical significance. Cochran’s Q test also yielded statistical significance (Χ^2^(9) = 20.1, *p* = 0.017). The *I*^2^ = 55.3%, suggesting marginally moderate heterogeneity across the 10 analyzed studies. The funnel plot is provided in Figure 3. The funnel plot showed only slight asymmetry, and Egger’s regression model yielded an intercept coefficient that was not statistically significant (bias = −3.3, 95% C.I.: −0.8–7.3, *p* = 0.230). Therefore, no publication bias was present.

### 3.4. Results of Sensitivity Analysis

Sensitivity analyses determined if Hedges’ g scores would differ after the removal of a single selected study. Removing Farmer et al. [32] slightly increased the pooled effect score to Hedges’ g = 0.18 (95% C.I.: 0.10–0.26), but this increase was not statistically significant. No other noteworthy changes were observed. Table 2 reports the results of this analysis.

## 4. Discussion

The aim of this study was to examine the effectiveness of playground-based interventions to improve accelerometer-assessed MVPA and to identify common aspects of the interventions that yielded favorable effects. There was an observed small but statistically significant pooled intervention effect. Furthermore, findings revealed only moderate study heterogeneity. Interventions that were successful for increasing MVPA used multiple methods including the use of multicolored markings and a sporting playground zonal design. Longer-duration interventions tended to not be as effective as shorter-duration interventions. Since school-based playground interventions utilizing different designs are pertinent ecological targets for increasing MVPA, the findings of this study are relevant to researchers, policymakers, stakeholders, and health and physical educator teachers. Given the multidisciplinary benefits of MVPA in pediatric populations [1,2,3], the playground setting should be an area to promote activity before, during, or after school. Interpretation of these findings, their practical applications, and recommended future research directions are discussed further.

While this meta-analysis reported a statistically significant positive effect of playground-based interventions on MVPA, the effect was small in magnitude. Several reasons may have contributed to the observed small effect. The primary reason was the relatively small SMDs of all individual selected studies, where the range was from −0.16 to 0.29. Another pertinent reason was the observed negative effects of two studies included in this meta-analysis [32,33]. While most of the included studies yielded a positive effect of playground-based interventions on increasing MVPA among children, two of the selected studies that implemented a playground-based intervention revealed decreases in accelerometer-assessed MVPA [32,33]. Intervention duration may have contributed to the relatively weak influence of playground intervention on children’s MVPA. The two studies that found decreasing trends in MVPA employed a relatively long-term playground intervention duration (1 year and 2 years) [32,33]. Longer-duration programs may decrease novelty and physical activity enjoyment [37], which are potential mediators of school-based MVPA promotion [38]. Although the duration of intervention yielded overall positive effects, Baquet et al. [26] highlighted the influence of intervention duration on sedentary behavior and light physical activity. The authors reported that sedentary and light physical activity tended to increase and MVPA tended to decrease over time from intervention commencement to its ending, possibly reflecting an attenuated novelty effect and loss of enjoyment [26].

Another salient factor that may have contributed to the observed small, pooled effect for the selected interventions was the use of accelerometers to assess the dynamic movements observed in playground settings. Interestingly, both Farmer et al. [32] and Hamer et al. [33] commented that using accelerometers to assess physical activity may obscure the types of activities participated in; that is, playgrounds tend to promote activities that involve movement of both the upper and lower limbs. Playgrounds also promote activities that incorporate a significant amount of muscular strength and endurance, domains that are not captured in accelerometer assessment. It is possible that the overall small effects observed in this meta-analysis are in part due to the use of accelerometer assessed MVPA. Although accelerometers are considered criterion assessments of ambulatory physical activity, they do not capture upper body activity and/or activity relating to intense non-ambulatory muscular contraction, physical activity, and fitness domains that are often promoted during playground-based interventions. A change in seasons can also influence children’s physical activity levels for long-duration interventions [39]. Generally, deteriorating weather conditions (e.g., extreme temperatures, rain, snow, or ice) are linked to higher sedentary behaviors and lower physical activity levels in children [40]. Long-term interventions may include many days of deteriorating weather conditions, which may inhibit outside play. By contrast, short-term interventions may have less variability in weather patterns and may be implemented during seasons with favorable weather conditions.

Despite the moderate study heterogeneity between studies included in this meta-analysis, an overall significant pooled effect of the playground-based physical activity interventions was found. Heterogeneity in physical activity interventions is not uncommon and can be explained by several factors. Our meta-analysis presented diversity in intervention duration (10 days to 2 years) and sample size (12 to 840 children). Additionally, some studies included the monitoring of MVPA and sedentary time during only morning and afternoon recess, while some studies monitored MVPA and sedentary time from both morning and lunch recesses. It has been found that children spend longer times in MVPA during the lunch break due to a longer recess time [31].

The playground setting is potentially beneficial for both increasing MVPA as well as improving the overall health and well-being of pediatric populations [41]. Physical inactivity has been correlated with higher cardiometabolic risk, anxiety and depression, and poorer cognitive performance [42,43,44]. Playground interventions have the potential to promote MVPA in addition to promoting time children spend outdoors, which itself has health-promoting benefits [45]. The results of this meta-analysis provide empirical evidence for the potential of playground interventions to increase MVPA, but future research should devise strategies to increase the magnitude of the improvements.

It is important to identify ways in which playground interventions are successful in promoting MVPA in children. The results from this study suggest that a variety of strategies can effectively accomplish this goal. One common strategy found in several of the interventions that were included in this review was altering or adding playground markings and utilizing zonal design along with the markings. For example, Baquet et al. [26] used brightly colored paint to add fun trails, hopscotch, and ladders to the playground. Along with these markings, the intervention playground was divided into three color-coded areas that indicated the types of activities that might be performed in that area. Adding playground markings and utilizing zonal design represent relatively low-cost ways for schools to promote MVPA. Other successful strategies included in this review were adding small equipment and/or novel structures [28,31,35]. Future research on the effect of playground interventions on physical activity in children should further explore low-cost ways to promote physical activity to create strategies that are feasible for schools to employ.

The limitations of this meta-analysis need to be considered before the results can be generalized. First, different types of control/comparison groups were used in each intervention. Relative to the intervention groups throughout the duration of a study, eight control groups followed usual physical activity programs. Other researchers could not identify appropriate control groups [33], used nature-based orienteering [27], or did not provide funding for playground development [35]. Second, moderate heterogeneity was observed across studies, possibly due to the variability in control groups in addition to varying study sample characteristics and sizes, intervention strategies, and accelerometer assessment methods. Third, only studies using accelerometer assessed MVPA were included in this study. Inclusion of other assessment methods such as self-report and/or systematic observation may have altered the findings. Fourth, only MVPA was assessed. Findings may have changed if sedentary behavior and light physical activity were also included in the analysis. Fifth, studies included in this meta-analysis included those from the United Kingdom, Australia, France, Belgium, and New Zealand. It is unknown if the results generalize to other areas. Potential moderators of effect such as age, sex, and socioeconomic status were not explored and should be examined in future research. When the study pool on this topic becomes sufficiently large, subgroup analyses should be conducted according to the type of intervention, methods of administration, presence of co-administrations, and the presence of cointerventions. Subgroup analyses and meta-regressions were not performed in the current study because of the small group sample sizes. Finally, all the reviewed studies were within school settings; therefore, future studies should review the non-school-based interventions that did not make the current study’s inclusion criteria.

## 5. Conclusions

Results from this meta-analysis show that playground-based physical activity interventions have small positive effects on increasing accelerometer-assessed MVPA in pediatric populations. Strategies that were common in the successful interventions included the addition of playground markings, utilizing zonal design, adding small equipment and/or novel structures, and allowing free play. Future research should be conducted to add to the growing body of literature on effective strategies to promote MVPA in playgrounds and to identify the most effective ways to promote MVPA in this environment. Schools should consider adopting some of these low-cost strategies to promote MVPA during times when the playground is being used, especially within the context of comprehensive school programs including recess, before school, and after school [46,47].

## Figures and Tables

**Figure 1 ijerph-19-03445-f001:**
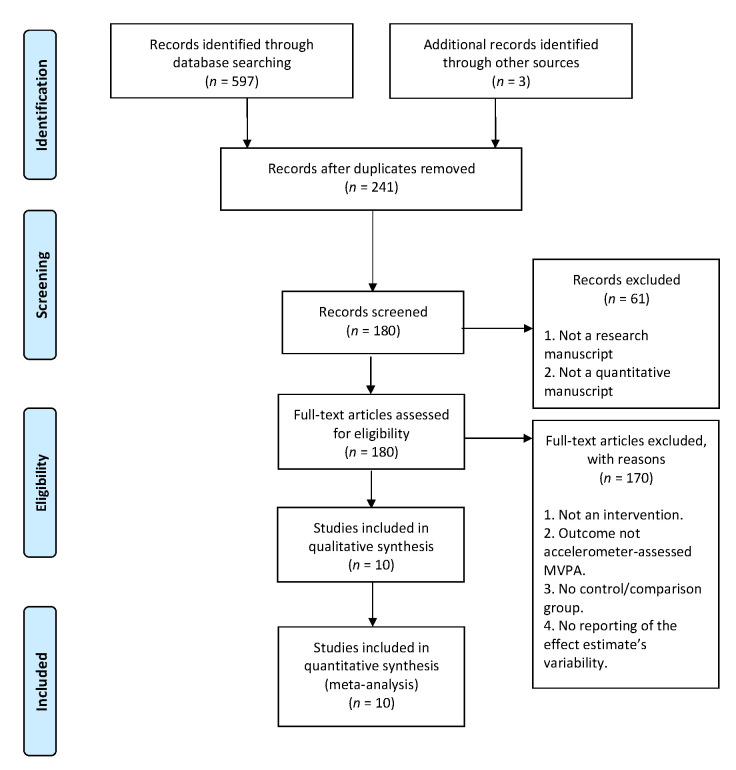
Flowchart of study extraction and inclusion.

**Figure 2 ijerph-19-03445-f002:**
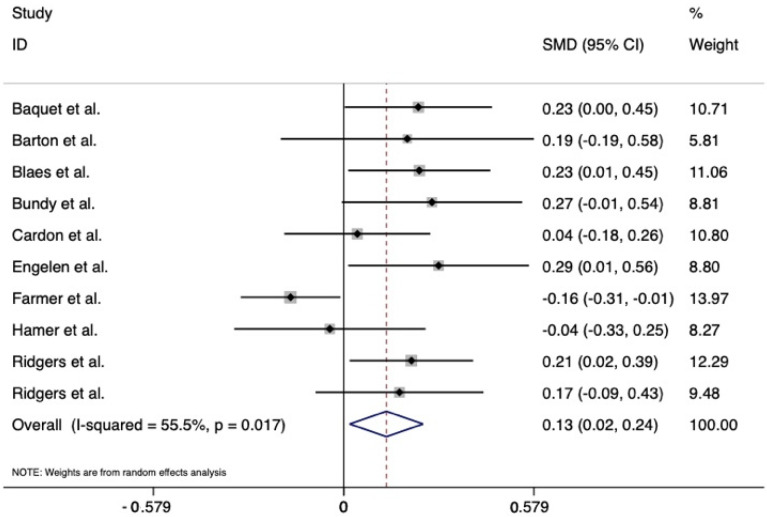
A forest plot showing the individual and pooled standardized mean differences. *Note:* SMD stands for standardized mean difference.

**Figure 3 ijerph-19-03445-f003:**
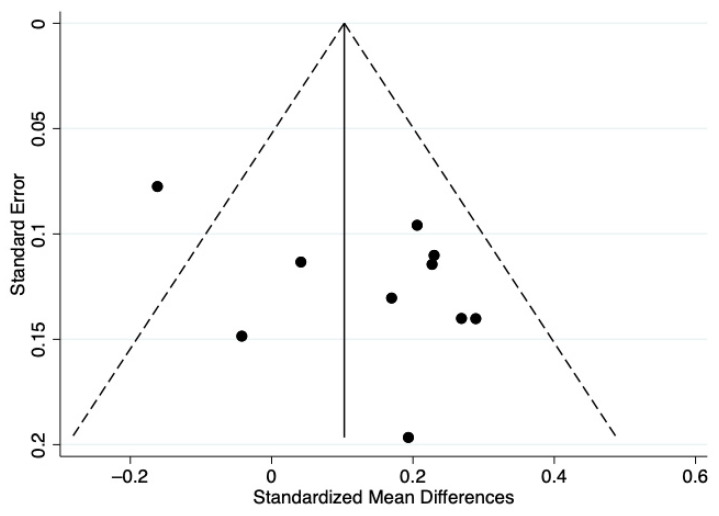
A funnel plot showing the association between standardized mean differences and standard error.

**Table 1 ijerph-19-03445-t001:** Summary of the extracted studies.

Study	Country	Age (years) (Mean ± SD)	*n*	Setting	Design	Duration	Accelerometer Processing	Other Outcomes
Baquet et al. [26]	France	IG: 8.3 ± 1.2 CG: 8.1 ± 1.8	283	3 Elementary schools	Playground markings; sporting playground zonal design	12 months	GT1 M ActiGraph Epoch set at 2 s Trost et al. cutpoints	BMI LPA SB
Barton et al. [27]	United Kingdom	8.8 ± 0.5	52	2 Primary schools	Small equipment; free play	5 days sports 5 days orienteering	GT1 M ActiGraph Epoch set at 1 s Treuth et al. cutpoints	Self-esteem
Blaes et al. [28]	France	IG: 8.7 ± 1.5 CG: 8.9 ± 1.6	332	4 Primary schools	Playground markings; sporting playground zonal design	4-day school week in April and May	GT1 M ActiGraph Epoch set at 2 s Trost et al. cutpoints	LPA SB
Bundy et al. [29]	Australia	6.0 ± 0.6	206	12 Primary schools	Recycled materials with no obvious play use; teacher/parent reframing workshop	13 weeks	GT3 X ActiGraph Epoch set at 5 s Evenson et al. cutpoints	Acceptance Self-competence SB Social interactions Social skills
Cardon et al. [30]	Belgium	5.3 ± 0.4	583	40 Pre-schools	Play equipment; playground markings	4–6 weeks	GT1 M ActiGraph Epoch set at 15 s Sirard et al. cutpoints	LPA SB
Engelen et al. [31]	Australia	6.0 ± 0.6	221	12 Primary schools	Recycled materials with no obvious play use; teacher-parent workshop	13 weeks	GT3 X ActiGraph Epoch set at 15 s Evenson et al. cutpoints	LPA SB
Farmer et al. [32]	New Zealand	IG: 7.9 ± 1.1 EG: 8.0 ± 1.2	840	16 Primary schools	Playground action plans	2 years	GT3 X ActiGraph Epoch set at 15 s Evenson et al. cutpoints	BMI BMI z-score Total PA Waist circumference Waist-to-height ratio
Hamer et al. [33]	United Kingdom	8.0	231	5 Primary schools 2 Secondary schools	Novel playground design based on emerging themes consultation	1 year	GT3 X ActiGraph Epoch set at 60 s Self-determined cutpoints (MVPA > 3000 cpm)	LPA SB
Ridgers et al. [34]	United Kingdom	IG: 8.3 ± 1.8 yrs CG: 8.0 ± 1.5 yrs	298	15 Primary schools	Sporting playground zonal design; novel structures; small equipment	6 weeks	Model 7164 ActiGraph Epoch set at 5 s Nilsson et al. cutpoints	HR-assessed MVPA VPA
Ridgers et al. [35]	United Kingdom	IG: 8.3 ± 1.8 CG: 8.0 ± 1.4	434	26 Elementary Schools	Playground markings; sporting playground zonal design; novel structures	1 year	Model 7164 ActiGraph Epoch set at 5 s Nilsson et al. cutpoints	HR-assessed MVPA HR-assessed VPA

IG stands for intervention group; CG stands for control group; MVPA stands for moderate-to-vigorous physical activity; cpm stands for counts per minute; BMI stands for body mass index; LPA stands for light physical activity; SB stands for sedentary behavior; HR stands for heart rate; VPA stands for vigorous physical activity.

**Table 2 ijerph-19-03445-t002:** Results from sensitivity analysis to examine changes in standardized mean differences by study removal.

Study Omitted	Adjusted Pooled SMD	Adjusted 95% CI
Baquet et al. [26]	0.12	0.00–0.23
Barton et al. [27]	0.12	0.01–0.25
Blaes et al. [28]	0.12	0.00–0.25
Bundy et al. [29]	0.12	0.00–0.24
Cardon et al. [30]	0.14	0.02–0.27
Engelen et al. [31]	0.11	0.00–0.22
Farmer et al. [32]	0.18	0.10–0.26
Hamer et al. [33]	0.14	0.03–0.25
Ridgers et al. [34]	0.12	0.00–0.25
Ridgers et al. [35]	0.13	0.01–0.24

SMD stands for standardized mean difference; 95% CI stands for 95% confidence interval.

## Data Availability

Data used in this study are available from each of the reviewed published papers.

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
