# Peer review of "The Effects of Playground Interventions on Accelerometer-Assessed Physical Activity in Pediatric Populations: A Meta-Analysis"

_ijerph, 2022, doi:10.3390/ijerph19063445_

Round 1

Reviewer 1 Report

The work has developed consistently with the set aims.
The meta analysis has been well-conducted and it is useful for the scientific
community dealing with children's health and well-being

The manuscript is clear and I think that it is useful for professionals dealing with the health and well-being of children. The references are mostly over the last 5 years, but this aspect does not affect the study. The experimental design is appropriate to gain aims of the study and the methods are punctually described, as well as the tables and the figures that properly show the data and make the understanding easy. The literature review was correctly done and the statistical analysis well applied. I believe that the article's area of strength is the correct application of the statistical tool. The limits are different, already stated by the authors themselves. I would suggest to consider a more conspicuous sample size, to extend the area of interest also to other Countries and to carry out a subgroup analysis on: criteria of inclusion / exclusion of participants, presence / absence of concomitant pathologies; type of intervention, methods of administration, presence of co-administrations, presence of co-interventions. Considering the school's prerogative to children’s education process, I would also envisage a more marked direction of meta-analysis towards the school environment to make the results of scientific researches more accessible to all professionals involved in health and wellness education.

Author Response

I would suggest to consider a more conspicuous sample size, to extend the area of interest also to other Countries and to carry out a subgroup analysis on: criteria of inclusion / exclusion of participants, presence / absence of concomitant pathologies; type of intervention, methods of administration, presence of co-administrations, presence of co-interventions. Considering the school's prerogative to children’s education process, I would also envisage a more marked direction of meta-analysis towards the school environment to make the results of scientific researches more accessible to all professionals involved in health and wellness education.

-Thank you for these insightful comments. Because the total study N was 10 – any subgroup analyses would have a very low subgroup N that will make sufficient power very difficult to achieve. This will also inflate type I error rate for conducting meta-analyses with low N. That being said, we agree with these comments and make suggestions for future research within the Discussion section. When the study pool will be larger to conduct these important subgroup analyses (lines 309-322).

Reviewer 2 Report

This is an interesting topic and the manuscript is well-written. 

I have minor suggestions throughout the manuscript.

  • Line 60: Change 'child' to 'children'
  • Line 81: What are the seven domains related to physical activity? I suggest to give examples.
  • Line 102: Inclusion criteria should be listed in a more organizing way. Were the articles that were playground-based intervention included? Were any alternative (e.g., nature-based interventions) included as well?
  • In Figure 1, include reasons excluded for 61 articles.
  • Line 139: Change 'fro' to 'for'
  • Table 1. Title row: I suggest either changing the second location to 'setting' or first location to 'Country'
  • Line 182: Spell check 'Remving'
  • Line 258-260: This sentence does not flow with the previous sentence. I suggest removing it. It is more suitable for introduction section.
  • Line 274-275: "Other successful strategies... free play" This sentence need reference.

Author Response

Line 60: Change 'child' to 'children'

- This has been changed in the new version of the manuscript (line 60).

Line 81: What are the seven domains related to physical activity? I suggest to give examples.

- We have listed out the seven domains related to physical activity in the manuscript (lines 82-83).

Line 102: Inclusion criteria should be listed in a more organizing way. Were the articles that were playground-based intervention included? Were any alternative (e.g., nature-based interventions) included as well?

- Excellent point. The inclusion criteria have been better organized and we have added more specifics to the criteria based on your suggestions.

In Figure 1, include reasons excluded for 61 articles.

-Thank you. These reasons now have been added within Figure 1.

Line 139: Change 'fro' to 'for'

- This has been changed in the new version of the manuscript (line 151).

Table 1. Title row: I suggest either changing the second location to 'setting' or first location to 'Country'

- Good point. We have changed Location to Country and Location to Setting (Table 1).

Line 182: Spell check 'Remving'

- This has been changed in the new version of the manuscript (line 198).

Line 258-260: This sentence does not flow with the previous sentence. I suggest removing it. It is more suitable for introduction section.

-This line has been taken out of the discussion section (line 275).

Line 274-275: "Other successful strategies... free play" This sentence need reference.

-Citations have been added (line 294).

Reviewer 3 Report

Overall, the study’s aims and findings align well with the aim and scope of the journal, and provide a useful contribution to the literature on playground modifications and MVPA in primary school settings. 

Author Response

Overall, the study’s aims and findings align well with the aim and scope of the journal, and provide a useful contribution to the literature on playground modifications and MVPA in primary school settings.

-Thank you for the kind and positive comments.